# ConBaT: Control Barrier Transformer for Safety-Critical Policy Learning

## Abstract

Large-scale self-supervised models have recently revolutionized our ability to perform a variety of tasks within the vision and language domains. However, using such models for autonomous systems is challenging because of safety requirements: besides executing correct actions, an autonomous agent needs to also avoid high cost and potentially fatal critical mistakes. Traditionally, self-supervised training mostly focuses on imitating previously observed behaviors, and the training demonstrations carry no notion of which behaviors should be explicitly avoided. In this work, we propose Control Barrier Transformer (ConBaT), an approach that learns safe behaviors from demonstrations in a self-supervised fashion. ConBaT is inspired by the concept of control barrier functions in control theory and uses a causal transformer that learns to predict safe robot actions autoregressively using a critic that requires minimal safety data labeling. During deployment, we employ a lightweight online optimization to find actions that can ensure future states lie within the safe set. We apply our approach to different simulated control tasks and show that our method results in safer control policies compared to other classical and learning-based methods.

## 1 Introduction

Mobile robots are finding increasing use in complex environments through tasks such as autonomous navigation, delivery, and inspection (Ning et al., 2021; Gillula et al., 2011). Any unsafe behavior such as collisions in the real world carries a great amount of risk while potentially resulting in catastrophic outcomes. Hence, robots are expected to execute their actions in a safe, reliable manner while achieving the desired tasks. Yet, learning safe behaviors such as navigation comes with several challenges. Primarily, notions of safety are often indirect and only implicitly found in datasets, as it is customary to show examples of optimal actions (what the robot should do) as opposed to giving examples of failures (what to avoid). In fact, defining explicit safety criteria in most real-world scenarios is a complex task and requires deep domain knowledge (Gressenbuch & Althoff, 2021; Braga et al., 2021; Kreutzmann et al., 2013). In addition, learning algorithms can struggle to directly infer safety constructs from high-dimensional observations, as most robots do not operate with global ground truth state information.

We find examples of safe navigation using both classical and learning-based methods. Classical methods often rely on carefully crafted models and safety constraints expressed as optimization problems, and require expensive tuning of parameters for each scenario (Zhou et al., 2017; Van den Berg et al., 2008; Trautman & Krause, 2010). The challenges in translating safety definitions into rules make it challenging to deploy classical methods in complex settings. The mathematical structure of such planners can also make them prone to adversarial attacks (Vemprala & Kapoor, 2021).

Within the domain of safe learning-based approaches we see instances of reinforcement and imitation learning leveraging safety methods (Brunke et al., 2022; Turchetta et al., 2020), and also learning applied towards reachability analysis and control barrier functions (Herbert et al., 2021; Luo et al., 2022). The application of learning-based approaches to safe navigation is significantly hindered by the fact that while expert demonstrations may reveal one way to solve a certain task, they do not often reflect which types of unsafe behaviors should be avoided by the agent. We can draw similarities and differences with other domains: natural language (NL) and vision models can learn how to generate grammatically correct text or temporally consistent future image frames by

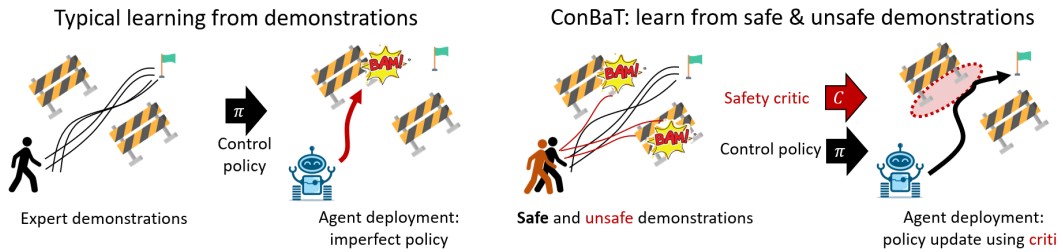

Figure 1: (Left) An agent trained to imitate expert demonstrations may just focus on the end result of the task without explicit notions of safety. (Right) Our proposed method ConBaT learns a safety critic on top of the control policy and uses the critic's control barrier to actively optimize the policy for safe actions.

following patterns consistent with the training corpus. However, for control tasks, notions of safety are less evident from demonstrations. While the cost of a mistake is not fatal in NL and vision, when it comes to autonomous navigation we find that states that disobey the safety rules can have significant negative consequences to a physical system. Within this context, our paper aims to take a step toward answering a fundamental question: how can we use agent demonstrations to learn a policy that is both effective for the desired task and also respects safety-critical constraints?

Recently, the success of large language models (Vaswani et al., 2017; Brown et al., 2020) has inspired the development of a class of Transformer-based models for decision-making which uses auto-regressive losses over sequences of demonstrated state and action pairs (Reed et al., 2022; Bonatti et al., 2022). While such models are able to learn task-specific policies from expert data, they lack a clear notion of safety and are unable to explicitly avoid unsafe actions. Our work builds upon this paradigm of large autoregressive Transformers applied on perception-action sequences, and introduces methods to learn policies in a safety-critical fashion.

Our method, named Control Barrier Transformer (ConBaT), takes inspiration in barrier functions from control theory (Ames et al., 2019). Our architecture consists of a causal Transformer augmented with a safety critic that implicitly models a control barrier function to evaluate states for safety. Instead of relying on a complex set of hand-defined safety rules, our proposed critic only requires a binary label of whether a certain demonstration is deemed to be safe or unsafe. This control barrier critic then learns to map observations to a continuous safety score, inferring safety constraints in a self-supervised way. If a proposed action from the policy is deemed unsafe by the critic, ConBaT attempts to compute a better action that ensures the safety of reachable states. A lightweight optimization scheme operates on the critic values to minimally modify the proposed action and result in a safer alternative, a process inspired by optimal control methods and enabled by the fully differentiable fabric of the model. Unlike conventional formulations of the control barrier function, our critic operates in the embedding space of the transformer as opposed to ground truth states, making it applicable to a wide variety of systems. We list our contributions below:

- We propose the Control Barrier Transformer (ConBaT) architecture, built upon causal Transformers with the addition of a differentiable safety critic inspired by control barrier functions. Our model can be trained auto-regressively with state-action pairs and can be applied to safety-critical applications such as safe navigation. We present a loss formulation that enables the critic to map latent embeddings to a continuous safety value, and during deployment we couple the critic with a lightweight optimization scheme over possible actions to keep future states within the safe set.

- We apply our method to two simulated environments: a simplified F1 car simulator upon which we perform several analyses, and a simulated LiDAR-based vehicle navigation scenario. We compare our method with imitation learning and reinforcement learning baselines, as well as a classical model-predictive control method. We show that ConBaT results in relatively lower collision rates and longer safe trajectory lengths.

- We show that novel safety definitions (beyond collision avoidance) can be quickly learned by ConBaT with minimal data labeling. Using new demonstrations we can finetune the safety critic towards new constraints such as 'avoid driving in straight lines', and the adapted policy shifts towards the desired distribution.

## 2 METHODS

We formulate the problem of learning safe control from observations as a partially observable Markov decision process that involves the agent interacting with an environment to receive high-dimensional observations $o$, which are used to take continuous-valued actions $a$. For simplicity, we treat the observations as equivalent to states $s$ while noting that actual agent states are implicit and not evident from the data. We define a trajectory $\tau$ as a set of state-action tuples $(s_t, a_t)$ over a discrete-time finite horizon $t \in [0, T]$. We assume that we have access to demonstrations from two sets of trajectories: $\tau \in \Sigma_s$, which obey the desired safety constraints at all time steps, and $\tau \in \Sigma_u$, which lead to an unsafe terminal state. Our goal is to learn a safe policy $\pi_{\text{safe}} : s \to a$ that results in actions that mimic the action distribution from good demonstrations $\Sigma_s$, while avoiding sequences of actions that lead to the unsafe terminal states of $\Sigma_u$.

### 2.1 BASE ARCHITECTURE: PERCEPTION-ACTION CAUSAL TRANSFORMER

The base transformer architecture for our method is derived from the Perception-Action Causal Transformer (PACT) proposed in Bonatti et al. (2022). PACT is primarily focused on creating a pretrained representation that can be finetuned towards a diverse set of tasks for a mobile agent. Its pretraining stage uses a history of state-action pairs from expert demonstrations to autoregressively train both a world model and a policy network, using imitation learning for its training objectives. The main functions from PACT that we employ are:

- **Tokenizer**: The state and action tokenizers operate on raw high dimensional observation and action data, and learn to represent them as compact tokens: $T_s(s_t) \to s'_t$, $T_a(a_t) \to a'_t$, where $s', a' \in \mathbb{R}^d$ and $d$ is a fixed length of the token embedding. $T_s$ and $T_a$ are learned functions, and we detail their architectures when describing concrete problem settings in subsequent sections.

- **Causal Transformer**: A set of Transformer blocks $X$ operates upon a sequence of state and action tokens. A causal attention mask is applied so that the transformer learns a distribution over the token at index $i$ given the history of tokens from $[0, i - 1]$. The output of this module is a sequence of state and action embeddings, as $X(s'_0, a'_0, s'_1, a'_1, ..., s'_T, a'_T) \to (s^+_0, a^+_0, ...s^+_T, a^+_T)$, where $s^+_t, a^+_t \in \mathbb{R}^d$.

- **Policy model**: An action prediction head acts as a policy and operates on the output state embedding $s^+_t$ at a given timestep $t$ and predicts the appropriate action: $\pi(s^+_t) \to \hat{a}_t$.

- **World model**: This module operates on the state and action output embeddings from the current timestep, and predicts the next state embedding: $\phi(s^+_t, a^+_t) \to \hat{s}'_{t+1}$. This module functions as a regularizing mechanism during training, and can optionally be utilized during the safe policy roll-out as explained next.

### 2.2 CONTROL BARRIER TRANSFORMER

On top of the original PACT architecture, we introduce important modifications that allow ConBaT to generate safe actions. While the original policy head does not explicitly distinguish between safe and unsafe behavior, which could result in sub-optimal action predictions, we use our policy head as an action proposal network which is later refined by the critic.

#### 2.2.1 CONTROL BARRIER CRITIC

We augment the transformer backbone with two trainable critic modules, which predict safety scores for the current and future expected states. The first critic $C : s^+_t \to \hat{c}_t \in \mathbb{R}$ maps the current state embedding $s^+_t$ to a real-valued safety score $\hat{c}_t$. The second critic $C_f : (s^+_t, a^+_t) \to \hat{c}_{t+1} \in \mathbb{R}$ estimates the future state safety score $\hat{c}_{t+1}$ based on the current state and action embeddings $s^+_t, a^+_t$. Here $C_f$ can be interpreted as conjoining both a world model and safety critic within a single network. We display our full ConBaT architecture along with the prediction heads in Figure 2(a).

To train these critics, we draw inspiration from the control barrier functions from classical controls literature (Ames et al., 2019). We assume our system to be of the form $\dot{s} = f(s, a)$ with $f$ locally Lipschitz continuous, state $s \in \mathcal{S}$ and action $a \in \mathcal{A}$. Let us denote the safe set of states as $\mathcal{S}_s \subset \mathcal{S}$.

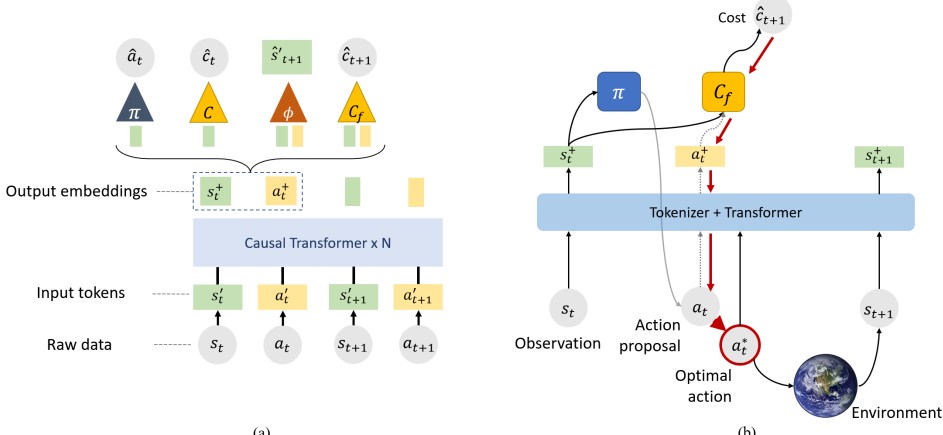

(a)

(b)

Figure 2: (a) The ConBaT architecture - a causal Transformer operates on state and action tokens $(s', a')$ to produce embeddings $(s^+, a^+)$. A policy head $\pi$ computes actions given state embeddings, and a current state critic $C$ computes a safety score. Both state and action embeddings are processed by a world model $\phi$ to compute the future state token, and by the future critic $C_f$ to produce a future safety score. (b) The deployment process for ConBaT involves a feedback loop. The future critic evaluates action proposals from the policy head to check safety of resultant states. The red arrows show the flow of gradients that allow optimizing for the safe action that results in a desired cost characteristic. The optimal action $a^*$ is used as the final command.

If there exists a function $h : \mathcal{S} \to \mathbb{R}$ and a policy $\pi^* : \mathcal{S} \to \mathcal{A}$ satisfying:

$$h(s) \geq 0, \forall s \in \mathcal{S}_s \tag{1}$$

$$h(s) < 0, \forall s \in \mathcal{S}_u = \mathcal{S}/\mathcal{S}_s \tag{2}$$

$$\dot{h}(s) = \partial h(s)/\partial s \cdot f(s, \pi^*(s)) \geq -\alpha h(s) \text{ with } \alpha > 0, \tag{3}$$

then $h$ is defined to be a control barrier function (CBF), and the policy $\pi^*$ ensures any initial state starting from $\mathcal{S}_s$ will always stay in $\mathcal{S}_s$ (forward invariant). The proof can be found in (Ames et al., 2014)(Theorem 1). Since $\mathcal{S}_s$ denotes the safe set, the policy $\pi^*$ will guarantee all the states initialized from the safe set to be always safe.

In practice, it is challenging to find the perfect safe policy $\pi^*$ for all states due to input constraints, imperfect policy learning, or disturbances (Qin et al., 2021; Dawson et al., 2022b). One alternative is to learn the CBF on top of an imperfect policy $\pi$ and then use an optimization routine (quadratic program (Ames et al., 2016), second-order cone program (Buch et al., 2021), gradient descent) to steer $\pi$ to satisfy CBF constraints, thus achieving safety. ConBaT follows a similar philosophy where the critics are learnt to approximate the CBF for a given (imperfect) policy $\pi$ (from PACT) so that the CBF conditions can be satisfied in most cases, and then use back-propagation to rectify $\pi$ to satisfy the CBF condition during deployment. In our method, the supervision signal needed to learn the critic values is just the binary labels indicating a state is safe or not, bypassing the need for more complex hand-designed signals. We call $C$ and $C_f$ Control Barrier-like critics (CBC) and detail the learning process below.

During CBC learning, instead of considering all possible state embeddings, we only enforce the state embeddings from the demonstrations $\Sigma_s$ and $\Sigma_u$ to satisfy the conditions above. In order for Equations (1)-(3) to hold, we expect the critic values to be positive on the collected safe state embeddings $\tilde{\mathcal{S}}_s^+$, negative on the collected unsafe state embeddings $\tilde{\mathcal{S}}_u^+$, and to not decrease too quickly for all collected state embeddings $\tilde{S}^+ = \tilde{S}_s^+ \cup \tilde{S}_u^+$. For $\tilde{S}_s^+$, we include all the states from the trajectories in $\Sigma_s$ and the states from the first $(L - 2T)$ time steps from the trajectories in $\Sigma_u$ where $L$ is the expert trajectory length and $T$ is the time horizon observed by the Transformer sequence. For $\tilde{S}_u^+$, we only consider the terminal states from the trajectories in $\Sigma_u$ (because those are the only 'unsafe' states we are certain). Note that $\tilde{S}_s^+$ and $\tilde{S}_u^+$ will be highly imbalanced, since we have more safe demonstrations than the unsafe ones, and also we only pick one state from each unsafe trajectory. But as shown empirically in future sections, we find out that the critic can be learnt effectively even with limited data using our formulation.

Training the CBC involves three loss terms. First, we employ a classification loss $\mathcal{L}_c$ to enable the CBC to learn the safe set boundary:

$$\mathcal{L}_c = \mathop{\mathbb{E}}_{s_t^+ \sim \tilde{\mathcal{S}}_s^+} \left[ \sigma_+ \left( \gamma - C(s_t^+) \right) \right] + \mathop{\mathbb{E}}_{s_t^+ \sim \tilde{\mathcal{S}}_u^+} \left[ \sigma_+ \left( \gamma + C(s_t^+) \right) \right] \tag{4}$$

where $\sigma_+(x) = \max(x, 0)$ and $\gamma$ is a margin factor that ensures numerical stability in training. The second loss enforces smoothness on the CBC values over time:

$$\mathcal{L}_s = \mathop{\mathbb{E}}_{s_t^+ \sim \tilde{\mathcal{S}}^+} \left[ \sigma_+ \left( (1 - \alpha)C(s_t^+) - C(s_{t+1}^+) \right) \right] \tag{5}$$

where $\alpha$ controls the local decay rate. Note that this loss is asymmetrical as it only penalizes fast score decays but permits instantaneous increases, as a fast-improving safety level does not pose a problem. The final loss ensures consistency between the predictions of both critics $C$ and $C_f$:

$$\mathcal{L}_f = \mathop{\mathbb{E}}_{s_t^+ \sim \tilde{\mathcal{S}}^+} \left[ \left| C_f(s_t^+, a_t^+) - C(s_{t+1}^+) \right| \right] \tag{6}$$

Theoretically, one could use a single critic $C$ coupled with a world model $\phi$ to generate $\phi(s^+, a^+) \rightarrow \hat{s}'_{t+1}$ and then estimate future CBC score as $C(\hat{s}'_{t+1})$. We found it empirically helpful to use a separate critic head $C_f$ to predict future CBC scores directly from the output embeddings, as it facilitates the action optimization process described in Section 2.2.2. The total training loss is $\mathcal{L}_{CB} = \lambda_c \mathcal{L}_c + \lambda_s \mathcal{L}_s + \lambda_f \mathcal{L}_f$, with relative weights $\lambda$.

### 2.2.2 OPTIMIZING ACTIONS TO IMPROVE SAFETY

As noted earlier, our goal is to learn a policy $\pi^* : s \rightarrow a$ that imitates the good demonstrations while always keeping the agent within the safe set. We compute $\pi^*$ through a two-step approach:

**Action proposal:** First, we use the output of the default prediction action head $\pi$ as a reasonable action proposal $\hat{a}$. This action is then propagated through the Transformer and the future state control barrier critic, and we compute the next state's safety score: $\hat{c}_{f,t+1} = C_f(s_t^+, \hat{a}_t^+)$.

**Action optimization:** If the future state is found to be in violation of the desired safety constraint, *i.e.* $\hat{c}_{f,t+1} < 0$, we need to override the default action so as to keep $s_{t+1}^+$ within the safe set $\mathcal{S}_s^+$. We intend to minimally modify the default action so as to result in a safe state, hence we express the optimal action as $\hat{a}_t + \Delta a^*$, and solve the following optimization problem for the modification $\Delta a$:

$$\Delta a^* = \mathop{\arg\min}_{\Delta a} \lambda ||\Delta a|| + \max(-C_f(s_t^+, \hat{a}_t^+ + \Delta a), 0) \tag{7}$$

Given the fully differentiable model, we can easily compute gradients of the cost with respect to the action through the control barrier function. We use a lightweight gradient descent optimization routine that results in the smallest action difference with respect to the original policy to keep the agent safe. We then take a step in the environment, collect new observations, and repeat the process. We show a visualization of the optimization step in Figure 2(b).

### 2.3 TRAINING PROCEDURE

We train ConBaT in a two-phase approach. Phase I is analogous to the original pretraining scheme for PACT (Bonatti et al., 2022), and focuses on training the policy head, world model, tokenizers and transformer blocks. From this phase, we aim to learn reasonable agent behavior from demonstrations. For phase II, we freeze the base network weights and add both control barrier critic modules and train with the combination of safe and unsafe demonstrations. By decoupling both training stages, we allow a user to potentially adapt a single base policy $\pi$ from a pretrained model towards different definitions of safety. Optionally, unsafe demonstrations can also be included in phase I to allow the world model to learn from the additional distribution of states, but the policy head is always trained only on safe samples. We empirically find that training the world model with data from unsafe demonstrations in the first phase results in a better final performance of the model.

## 3 EXPERIMENTAL RESULTS

We apply ConBaT to two domains in simulation, which we visualize in Figure 3:

**F1/10 race car.** We use a simulation of a racing car that navigates within multiple 2-D racing tracks (O'Kelly et al., 2020). The car receives observations at every step comprised of the distance and the angle relative to the center line. The action corresponds to the steering angle, and the goal is to learn how to drive safely without colliding with the track's edges. This is a toy scenario used to demonstrate the functionalities of our method, upon which we conduct several ablation studies;

**MuSHR car.** Inspired by the MuSHR car (Srinivasa et al., 2019), we use a vehicle simulation in a more realistic navigation setting. The car is equipped with a 2D range sensor, returns LiDAR scans as observations, and takes steering angles as actions. We use a 2D map scanned from a real office space of approximately $30 \times 70$ meters of area, for both data collection and safe navigation deployment without collisions.

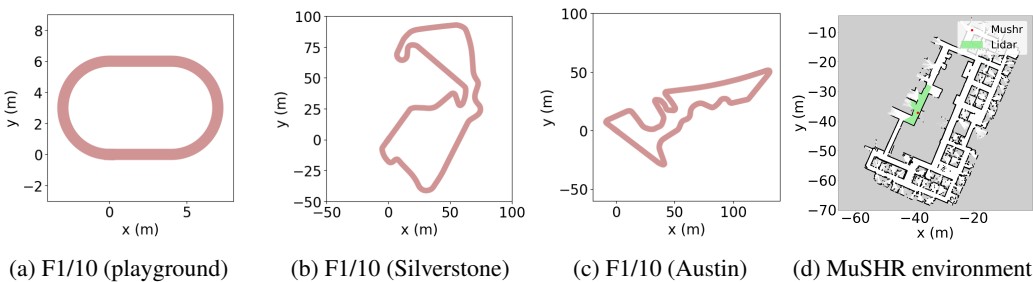

(a) F1/10 (playground)   (b) F1/10 (Silverstone)   (c) F1/10 (Austin)   (d) MuSHR environment

Figure 3: Simulation environment visualization.

We collect sets of safe and unsafe trajectories in both domains. Each trajectory has a discrete binary safety label, which is used to train the safety critic. Further details about the domains and data collection can be found in Appendix A.

**Metrics:** We evaluate our models in both domains by deploying the learned policies and evaluating the rollout trajectories. We measure the following metrics: (1) *collision rate*, measured as a percentage of trajectories in the test set that end in a crash within the cut-off time horizon; and (2) *average trajectory length* (ATL), which corresponds to the average length of deployment trajectories, expressed in number of time steps before crashing or time-out if no crash occurs.

### 3.1 SAFE NAVIGATION ANALYSIS

We first examine the safe navigation performance of ConBaT in the F1/10 simulator, and compare it with PACT. We train the models with 1K demonstrations, each 100 timesteps long, from the *Playground* track (Figure 3(a)). During deployment, we roll out 128 trajectories for each model for a maximum of 1000 timesteps. ConBaT achieves a 0% collision rate whereas PACT collides at some point in every instance.

Next, we apply the ConBaT model trained on *Playground* on more challenging tracks such as *Silverstone* and *Austin* (Figure 3(b),(c)). We compare our model's performance against a frozen PACT model (PACT - trained only on *Playground*) as well as against a version finetuned on these tracks (PACT-FT). In Table 1 we observe that ConBaT is able to learn the safety concepts effectively enough to generalize to new tracks, outperforming both PACT and PACT-FT. In Figure 4, we take a closer look at the performance on *Silverstone* and show a histogram of the trajectory lengths across the test dataset. A PACT model trained on *Playground* registers fairly low ATL values, whereas PACT-FT performs better, sometimes even reaching the maximum length of 1000. However ConBaT significantly outperforms both, with all the trajectories being collision-free.

### 3.2 COMPARISON WITH BASELINE METHODS

In the MuSHR simulation we collect 10K trajectories to create a dataset upon which we train different baseline safety models for comparison against ConBaT. We compare with baselines from three different categories: imitation learning (IL), for which we use a naive behavior cloning approach

|            | PACT | PACT-FT | ConBaT |
|------------|------|---------|--------|
| Playground | 100  | -       | **0.0** |
| Silverstone | 100 | 96.88   | **0.0** |
| Austin     | 100  | 100     | **61.7** |

(a) Collision Rate (%) - lower is better

|            | PACT  | PACT-FT | ConBaT |
|------------|-------|---------|--------|
| Playground | 175.45 | -      | **1000** |
| Silverstone | 61.57 | 439.28 | **1000** |
| Austin     | 57.11 | 165.12  | **678.14** |

(b) Avg. Trajectory Length - higher better

Table 1: Comparison of PACT and ConBaT for the F1/10 task. ConBaT outperforms PACT

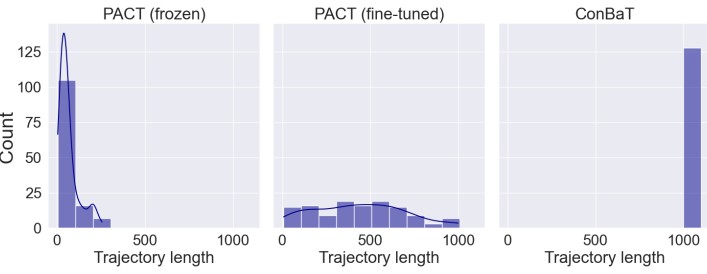

Figure 4: Histogram of trajectory lengths in the Silverstone track: ConBaT generalizes efficiently to a new track, achieving a 100% success rate, whereas both frozen and finetuned PACT fall short.

(BC), PACT, and Generative Adversarial Imitation Learning (GAIL) (Ho & Ermon, 2016); reinforcement learning, for which we evaluate PPO (Schulman et al., 2017), TRPO (Schulman et al., 2015), and SAC (Haarnoja et al., 2018). We also implement a model predictive control baseline via the CasADi solver (Andersson et al., In Press, 2018). Further details about the implementations of these baselines can be found in Appendix B. During simulation deployment, we run 128 trajectories for each model, each for a maximum of 5000 timesteps. Figure 5 shows ConBaT compared against all baselines. Even on this complex setting using high-dimensional LiDAR observations in a realistic map, we observe that ConBaT achieves the lowest collision rate and the highest ATL.

## 3.3 LEARNING A NEW SAFETY DEFINITION

An important feature for a safety critic is to be able to learn different definitions of safety without major architectural changes or hand-crafting. Therefore we evaluate ConBaT's ability to learn new safety constraints beyond simple notions of collision-free navigation. We consider the hypothetical situation of a vehicle agent where traveling in a straight line is undesirable, and only curved motions are allowed (due to dynamical constraints, or perhaps to impress other agents in the environment). We generate a new dataset with unsafe labels for every straight trajectory segment, and train a new critic which we call ConBaT-NS (not-straight).

As seen in Figure 6, ConBaT-NS succeeds in moving the vehicle through curved trajectories only, even in straight hallways, showing that new constraints can be incorporated with ease. Figure 6c also displays how this new constraint is reflected in the policy's action distribution output. We note that both ConBaT and ConBaT-NS are finetuned over the same base PACT model, which demonstrates the potential for training multiple critics mapping to distinct safety constraints.

## 3.4 ABLATION STUDIES

**Variations in critic architecture:** As discussed in Section 2, the control barrier critic module could be architected in different ways. We compare different architectural choices for the critic:
*CBC-NW*: Both $C$ and $C_f$ are used, but there is no world model in the architecture.
*CBC-CW*: Only the current state critic $C$ is used, and the output of the world model is fed into it for the next state's cost.
*CBC-TF*: All the modules $C$, $C_f$, and $\phi$ are used. The input to $C_f$ is the output state embedding and the action input token $(s_t^+, a_t')$.
*CBC-EF*: All the modules $C$, $C_f$, and $\phi$ are used. The input to $C_f$ is the output state embedding and the output action embedding $(s_t^+, a_t^+)$.

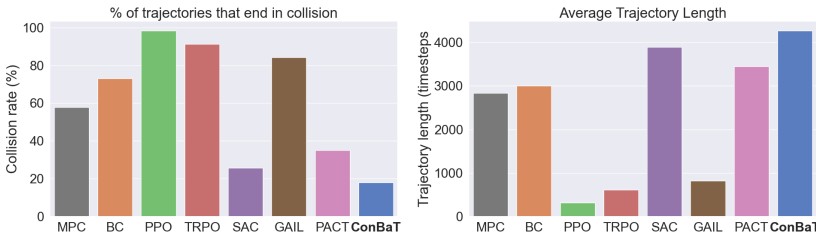

Figure 5: ConBaT outperforms classical MPC and several learning-based methods on safe navigation in the 2D MuSHR car domain.

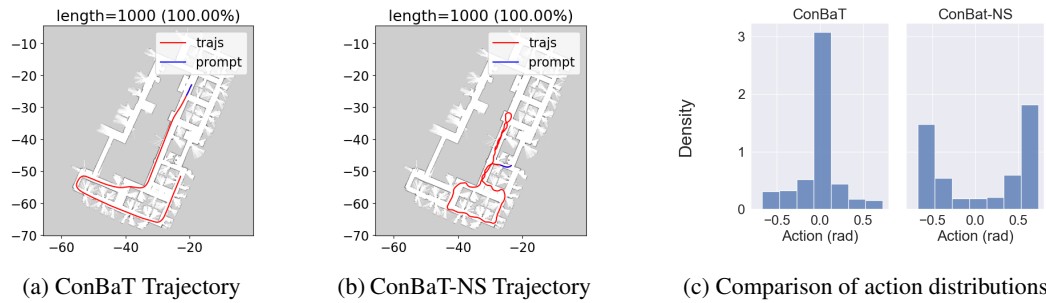

(a) ConBaT Trajectory      (b) ConBaT-NS Trajectory      (c) Comparison of action distributions

Figure 6: Demonstration of ConBaT being used for a new safety constraint where actions mapping to the vehicle going straight are deemed unsafe.

We compare all methods in Table 2a. We note that the world model is indeed useful, as evidenced by the lowest performance of CBC-NW. Using a future state critic results in slightly better performance than only using a current state critic that takes world model simulated states, but higher computational effort. Lastly, CBC-EF achieves higher performance than CBC-TF, indicating that the critic benefits from the context that plays a part in the computation of the action embedding, as opposed to the single token. The CBC-EF architecture is used for all the other experiments in the paper.

| Arch | Collision (%) | ATL (# steps) | Runtime (s) |
|------|---------------|---------------|-------------|
| PACT | 100 | 175.45 | 63.328 |
| CBC-NW | 4.69 | 952.47 | 129.595 |
| CBC-CW | 1.56 | 983.48 | 200.153 |
| CBC-TF | 3.91 | 972.42 | 78.159 |
| CBC-EF | 0.78 | 991.22 | 130.032 |

(a) Different architectures for online optimization.

| % Trajs | Collision (%) | ATL (# steps) | Runtime (s) |
|---------|---------------|---------------|-------------|
| 5 | 5.47 | 944.67 | 120.427 |
| 10 | 3.91 | 960.18 | 132.079 |
| 25 | 1.56 | 983.47 | 130.049 |
| 50 | 3.91 | 960.26 | 127.275 |
| 100 | 0.78 | 991.22 | 127.073 |

(b) Amount of unsafe training trajectories used.

Table 2: Ablation study over critic architecture and amount of unsafe trajectories needed for Phase II training

**Data Requirements:** The existence of demonstrations that include unsafe terminal states is critical for ConBaT to be able to encode the safety constraints. Yet, given that simulating and collecting a large amount of unsafe demonstrations can be challenging (and costly in real world conditions), we perform a analysis of how ConBaT performs in low-data regimes when training the critic (phase II). Table 2b reports performance levels for varying unsafe dataset sizes. We observe that ConBaT is effective at achieving reasonable safe navigation even with small amounts of unsafe demonstrations.

## 4 BACKGROUND

**Learning Policies from Data:** Imitation Learning (IL) is a popular method for policy learning, and requires expert demonstrations (Schaal, 1996; Atkeson & Schaal, 1997; Pastor et al., 2009; Behbahani et al., 2019; Billard & Grollman, 2013; Ekvall & Kragic, 2008). The most naive IL approach is behavioral cloning (BC) (Bain & Sammut, 1995), which directly learns the mapping from states to actions using supervised learning from expert data. Such simple applications of IL cannot generalize

to the out-of-distribution scenarios induced by on-policy deployment, and often requires additional training routines (Ross et al., 2011). Inverse reinforcement learning (IRL) (Ng et al., 2000; Natarajan et al., 2010) can mitigate this challenge, as it seeks to recover the original expert's cost function. Alternatively, one may use generative adversarial imitation learning (GAIL) (Ho & Ermon, 2016; Bhattacharyya et al., 2018), which employs a discriminator to differentiate an expert's policy from other behaviors generated by a policy network. When it comes to safety, however, none of the above methods incorporate the concept of safety directly into their learning procedures.

Reinforcement learning can enable safe policy learning through rewards that not only encourage good behaviors but also penalize unsafe states. Bhattacharyya et al. (2019) use an augmented reward for traffic simulation to guide safe imitation, and we find numerous examples of safe policy design (Liu et al., 2020; Berkenkamp et al., 2017; Cheng et al., 2019; Li & Belta, 2019). However, manual reward shaping is laborious and nontrivial, often leading to undesired consequences. In comparison, our approach with learned control barrier functions uses minimal safety labels.

**Safe policy learning using control barrier functions:** Control barrier function (CBF) is a classical concept to ensure system safety (Prajna & Jadbabaie, 2004; Wieland & Allgöwer, 2007; Ames et al., 2014; 2019). Recently, several works have demonstrated learning safe policies based on existing CBFs (Chen et al., 2017; Borrmann et al., 2015), constructing CBF jointly with the policy learning via Sum-of-Squares (Wang et al., 2018), leveraging SVMs (Srinivasan et al., 2020) or Neural Networks (Ferlez et al., 2020; Qin et al., 2021; Meng et al., 2021; Dawson et al., 2022a;b). However, these methods often assume access to raw state inputs and ground truth safety labels from the environment. Instead, ConBaT works on the embedding space akin to models that plan using imagination (Okada & Taniguchi, 2021; Wu et al., 2022), learns purely from offline data, and employs an online policy rectification process to achieve safety.

**Safe policy learning with predictive world models:** ConBaT is aligned with the Mode-2 proposal in LeCun (2022), where the agent makes action proposals, evaluates the costs from future predictions, and then plans for the next action. Predicting the next state based on the current state and action is fundamental to model predictive control (Bryson & Ho, 2018). The idea of predicting the future costs via the world model can be traced back to (Schmidhuber, 1990). When it comes to high-dimension state/observation space, latent embedding dynamics are learned to help the agent to achieve high performance in reinforcement learning (Okada & Taniguchi, 2021; Wu et al., 2022), and sequence predictions (Giuliari et al., 2021; Chen et al., 2021; Micheli et al., 2022). In our case, we learn a predictive critic on the embedding space with CBF conditions as guidance.

## 5 CONCLUSIONS

In this work, we propose ConBaT, a framework that learns safe navigation directly from demonstrations. Our method leverages causal Transformers coupled with a differentiable safety critic that is inspired by control barrier functions in control theory. The control barrier critic module implicitly builds a safe set for states from discrete safety labels, bypassing the need for complex mathematical formulations of safety constraints. We apply our method to two simulated domains and show that our method outperforms existing classical and learning-based safe navigation approaches. Furthermore, we show that ConBaT can be quickly adapted to new safety constraints from limited demonstrations. This paradigm of learning safety constraints implicitly from minimal labels reduces training effort in creating safe agents and makes it easier to adapt to new definitions of safety.

Though empirically ConBaT can learn safer policy and quickly adapts to new safety concepts, there are some limitations. Our method might not guarantee safety if ConBaT makes wrong CBF score predictions due to out-of-distribution data, or the online optimization falls into a local minimum. One solution is to follow a paradigm like DAGGER (Ross et al., 2011) to keep exploring the environment and collect new states and labels to update our CBF critics and world model. The runtime for ConBaT in evaluation is also $0.2 \sim 1.0x$ higher than other learning-based methods due to the back-propagation step in online optimization. We discuss different ConBaT architectures in Sec. 3.4 to weigh the trade-off between the safety performance and the runtime. Further runtime improvement can be made by faster back-propagation algorithms optimized for GPU hardware. Currently, our framework was only applied in simulated environments. In the future, we wish to extend our approach to image-based control in 3D and toward more complex robotic platforms in simulation and the real world.

## 6 ETHICS STATEMENT

ConBaT is a tool that learns safety concepts with a neural approximation of a control barrier function. Given the finite nature of the datasets used to train our model, and despite performing an optimization routine on top of the safety critic's judgement, we may still encounter failure models if the critic imperfectly approximates the safety function. Therefore, ConBaT may give users a false sense of security during deployment. We recommend the use of backup methods such as emergency maneuvers if ConBaT is used in safety-critical real-world applications, besides extensive testing in simulation and controlled realistic scenarios. Additionally, since ConBaT reproduces patterns found in its training datasets that contain safe and unsafe trajectories, it may reproduce data biases during deployment.

## 7 REPRODUCIBILITY STATEMENT

We are working towards providing an open-sourced implementation of our datasets, simulators, models and evaluation framework in the near future. We will make the links available for the camera-ready version of this work.

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
