# OpenReview forum: "ConBaT: Control Barrier Transformer for Safety-Critical Policy Learning"
_ICLR.cc/2023/Conference — Submitted to ICLR 2023_

### Official Review · Reviewer_okow · 2022-10-19

**Confidence:** 3
**Correctness:** 4
**Technical Novelty And Significance:** 3
**Empirical Novelty And Significance:** 2
**Recommendation:** 5

**Clarity, Quality, Novelty And Reproducibility:**

I see that information about the baselines are in the appendix, but I believe they could be in the main paper. The experiment in the MuSHR environment is arguably the most important in the paper, but very little detail is afforded to it.

Otherwise the quality and clarity of this paper is very good. The novelty seems good to me (although I'll admit I'm not as familiar with the safety literature). The reproducibility is pretty good--hyperparameters and training details are given in the appendix--but could be improved by including code in the supplemental material.

**Strength And Weaknesses:**

Strengths:

The idea is pretty clever, and very well motivated by control barrier functions. In addition, I quite enjoyed reading about the ConBaT-NS experiment.

Weaknesses:

The biggest weakness is the lack of experiments, and the fact that they are unconvincing.

The first experiment compares ConBaT to PACT and PACT-FT. This experiment seems more of a proof of concept, as neither PACT nor PACT-FT are given access to safe/unsafe trajectory labels (as far as I can tell). Of course ConBaT should outperform PACT here.

The second experiment compares ConBaT to a variety of imitation learning and RL baselines. The number of baselines is high, but their implementation is not clear to me. From the appendix I see that the RL baselines have a hand-designed reward function discouraging collisions. This makes comparing ConBaT to the RL baselines difficult as one is given a reward function and one is given expert demonstrations with safe/unsafe labels. The imitation learning baselines also do not seem to get access to the the safe/unsafe labels (but please correct me if I'm wrong), making comparison difficult. What this paper could benefit from is baselines that are more comparable to ConBaT's setup.

Some other things I think this paper could benefit from:

Analysis of how often and when the CBF conditions are satisfied by the critics. This could be interesting to look at: for example, see if unsafe regions really do have negative safety score, or how consistent the critics are in different regions.

On a related note, are all the CBF conditions necessary? Do we need to optimize all three losses? ConBaT may be inspired by CBFs but perhaps the method could be simplified.

An aside: I don't quite understand the emphasis on the transformer aspect of this method. As far as I can tell these critics can be trained on a variety of architectures, and emphasizing transformers detracts from the core idea. Perhaps this versatility could be a benefit.

Another aside: I don't understand why the RL baselines don't just learn to drive in circles. Does the turning radius prevent this?

**Summary Of The Paper:**

The authors propose to learn a "control barrier critic" (CBC) on top of PACT, an autoregressive world model and policy trained through imitation learning. The CBC is loosely inspired by control barrier functions (CBFs), a scalar function of state which if can be found for a system and policy guarantees that if the policy starts in a safe state will always stay in the set of safe states. The CBC is trained using expert demonstrations along with safe/unsafe labels, and is optimized to satisfy the conditions of true CBFs.

Once a CBC (safety score) has been optimized, it can be used to improve the safety of the policy at inference time. Because the safety score should be positive, if an action taken by the policy were to achieve a negative safety score a minimum perturbation could be applied to that action to ensure the safety score is non-negative in the next time-step. This minimum perturbation is found through gradient descent.

The authors present experiments in two simulated environments: F1/10, a 2D racing simulation, and MuSHR, a more complex simulation of a toy car navigating an office space with LIDAR. ConBaT is shown to be safer than PACT or finetuned PACT, and beats various baselines on the MuSHR environment.

**Summary Of The Review:**

The paper is well written, and the idea is novel and well motivated. However, the lack of convincing experiments concerns me (see "weaknesses" section). It is hard to compare ConBaT against baselines, and as a result it's not clear if the method is actually effective.

---

### Official Review · Reviewer_89Hu · 2022-10-23

**Confidence:** 4
**Correctness:** 3
**Technical Novelty And Significance:** 2
**Empirical Novelty And Significance:** 3
**Recommendation:** 6

**Clarity, Quality, Novelty And Reproducibility:**

As mentioned in the text, control barrier functions are being used widely in learning-based control techniques, which limits the novelty of this work. The paper is clearly written, and experiments seem to be decent, with some interesting ablations. The core novelty of this work is the use of the learned safety critics within a transformer based policy.

**Strength And Weaknesses:**

Strengths:

The idea makes sense, the paper is well written and motivated, and the proposal to use two critics is an interesting one, with ablations  exploring this valuable.

Weaknesses:

Baselines. As far as I can tell, none of these baselines consider "safe" policy learning, so while the comparison is useful to show that the proposed approach works as well as those that don't consider safety, I feel it would be valuable to put this into perspective with some of the approaches in Brunke 22 or those mentioned in related work.

It is also unclear to me whether these approaches had the benefit of the additional safe/unsafe training data, which would presumably be quite informative to the policy. It looks like they don't get this information, which does not really make the comparison very fair?

The proposed approach also needs unsafe demonstrations to learn, which isn't always practical (i.e. I didn't learn to drive by crashing a lot and being told that sequence of states and actions was "bad"). Is it fair to term this a "safety-critical policy learning" approach, if it needs to violate safety constraints to learn? As I understand it the resultant policy may be safe, but to me the primary purpose of a barrier function is to prevent any violation of the safety constraints, both during learning and after so I found the link to control barrier functions a little tenuous.

Ablations show that relatively large amounts unsafe data would be required to obtain an acceptable safety level.

Along these lines, I think the term "safe policy learning"  is ambiguious and potentially misleading (although I recognise the precedent in this field). The key distinction is that the learning process is clearly not safe, although the resultant policy may be safe(-ish). I think this needs improvement (eg. learning safe policies)

**Summary Of The Paper:**

This paper introduces a method for safe policy learning that relies on a causal transformer architecture that proposes actions, and a pair of critics that predict a current safety score and a future safety value. The method is inspired by control barrier functions, and critics trained using binary supervision (safe and unsafe trajectories.) Results on two simulated navigation environments show that the method is effective and outperforms baselines that (do not?) consider safe learning.

**Summary Of The Review:**

This paper proposes an interesting model and approach to learn safe policies, so I recommend acceptance. However, this approach is only applicable in certain situations where examples of unsafe and safe trajectories can be obtained, which limits the applicability of this work somewhat. I'd also value some more precise language to avoid giving readers the impression that this approach allows for safe policy learning.

---

### Official Review · Reviewer_iZzR · 2022-10-24

**Confidence:** 4
**Clarity, Quality, Novelty And Reproducibility:** See above
**Correctness:** 3
**Technical Novelty And Significance:** 3
**Empirical Novelty And Significance:** 2
**Recommendation:** 5

**Strength And Weaknesses:**

## Paper strengths and contributions
**Motivation and intuition**
The motivation for studying safety-critical imitation learning is convincing.

**Novelty**
Considering safety constraints for imitation learning problems is intuitive and convincing. This paper presents an effective way to implement this idea.

**Technical contribution**
Learning the Control Barrier-like critics (CBC) to predict the value of safety seems effective, especially when the unsafe annotated trajectories are available for training.

**Clarity**
- The overall writing is clear. The authors utilize figures well to illustrate the ideas. Figure 1 clearly shows the scenario of the safety-critical imitation learning problem.
- The paper gives clear descriptions in intuitive ways. The notations, formulations, and designs of the learning framework are well-explained.

**Related work**
The proposed method is based on prior work (i.e., Perception-Action Causal Transformer). The authors clearly describe the prior work and how they extend it to address the safety-critical problem. Also, sufficient relevant literature is discussed to help readers understand the context.

**Reproducibility**
The paper states that the authors are working towards providing an open-sourced implementation of the proposed method.

## Paper weaknesses and questions
**Trade-off between safety and performance**
Safety-critical policy learning is mentioned as a strength in the paper, but this comes at the cost of performance. For a car agent that aims to reach a goal from a starting point, it may choose a safer but longer path with the proposed algorithm. The experiments use the collision rate and the average trajectory length as metrics, so the policy only needs to ensure no crash occurs. I would also like to see experiments that consider the policy's optimality.

**Experiment**

- The description of the experimental setups is not clear to me. While the claims seem to be promising, the amount of unsafe training trajectories is not provided, which is very important to determine how practical the proposed method is. To be more specific, for the F1/10 race car environment, the authors train the model with 1K demonstrations (Sec 3.1) where the expert trajectory is generated from a search-based model predictive control (Supp. A.1). How many trajectories are unsafe among the 1K training samples? How to generate and collect such unsafe trajectories? The statistic of the training data is not specified for the MuSHR environment.
- The comparisons to the previous works may be unfair. First, comparing the proposed method with reinforcement learning methods such as PPO, TRPO, and SAC, is unreasonable. The proposed method learns from demonstrations generated by a search-based expert model. In contrast, the RL algorithms learn from a pre-defined reward function (Supp. B.2). Comparing methods with different input signals is improper. Second, the authors use the same expert trajectories containing unsafe samples for ConBat and other imitation learning algorithms. The imitation learning algorithms such as BC and GAIL learn from the expert data, so inferior results during deployment are expected. Since the annotations of the trajectories are available, it is more reasonable to use collision-free samples to learn those imitation learning baselines.

**Summary Of The Paper:**

This paper addresses the problem of safety-critical imitation learning, where the learned policy needs to avoid unsafe actions. To this end, the paper builds on the Perception-Action Causal Transformer and augments the framework to learn from safe and unsafe demonstrations. While the experiments show that the derived policy performs reasonably well in two simulation environments, the experimental setups are unclear and the ablation study is missing. I believe this work extends previous work to a practical safety-critical scenario, but the experiments must be improved to support the proposed method.


**Summary Of The Review:**

The author introduces a convincing learning framework for a practical safety-critical imitation learning problem. However, the experimental setup is unclear and the ablation study is missing. Also, the comparisons to the related works are not entirely fair. In sum, I am leaning toward rejecting this paper because the provided experimental results are not solid enough to support the effectiveness of the proposed method.

---

### Official Review · Reviewer_tYu8 · 2022-11-01

**Confidence:** 3
**Correctness:** 2
**Technical Novelty And Significance:** 2
**Empirical Novelty And Significance:** 1
**Recommendation:** 3

**Clarity, Quality, Novelty And Reproducibility:**

Paper is written with clarity.
The proposed method is incrementally novel in comparison with the baseline method PACT but isn't conceptually or technically novel in general. More related work should be included as baselines.

**Strength And Weaknesses:**

This paper aims to build safe systems, which is important for learning-based models. The paper motivated the problem by emphasizing the criticalness of ensuring safety for autonomous agents but the proposed method does not guarantee safe behavior. While the method is claimed to be inspired by control barrier functions, the proposed method just use failed demonstrations as examples of states/state-actions with negative values (assigned by the critic). No benefit of control barrier functions is actually seen here. At the same time, there are many RL methods that leverage failed demonstrations or learn from failed experiences but there is no such baseline compared in the experiments.

**Summary Of The Paper:**

The paper propose a method to learn from both safe and unsafe demonstrations by constructing a critic that assigns positive values to safe demonstrations and negative values to unsafe demonstrations. The proposed model leverages a transformer backbone. Experiments in simulated environments demonstrate the proposed method outperforms the baseline that does not leverage failed demonstrations.

**Summary Of The Review:**

This paper proposed an incrementally novel idea with little conceptual novelty. Experiments are limited in simulated domain and only one weak baseline is included. I do not recommend accepting this paper.

---

### Decision · Program_Chairs · 2023-01-20

**Decision:**

Reject

**Justification For Why Not Higher Score:**

Reviewers are skeptical about the presented experiments and baseline comparisons. Real benefits of the algorithm are unclear.

**Justification For Why Not Lower Score:**

N/A

**Metareview: Summary, Strengths And Weaknesses:**

The paper uses an transformer architecture to learn from demonstrations labelled as safe and unsafe.It uses a pair of critics that predict a current safety score and a future safety value and a trained with a binary classification loss. Experiments in simulation demonstrate the performance of the proposed method

The paper contains an interesting model to learn from safe and failed demonstrations. Yet, most reviewers were concerned about the experiments as it was hard to compare the proposed algorithms fairly to the given baselines. While the authors response could clarify some of these issues, 2 reviewers did still see the experiments as main issue and did not change their score. Reviewer tYu8 was unfortunately unresponsive during the discussion and his concerns seem to be addressed adequately, hence; I give less weight on his review. Yet, as the reviewers that did respond are still skeptical about the paper, I encourage that the authors work on their experiments to better show the contribution of their approach. I further recommend that the paper has to go through another review proces.

**Summary Of Ac-Reviewer Meeting:**

N/A